# Annexin A1 as Neuroprotective Determinant for Blood-Brain Barrier Integrity in Neonatal Hypoxic-Ischemic Encephalopathy

**DOI:** 10.3390/jcm8020137

**Published:** 2019-01-24

**Authors:** Ruth Gussenhoven, Luise Klein, Daan R. M. G. Ophelders, Denise H. J. Habets, Bernd Giebel, Boris W. Kramer, Leon J. Schurgers, Chris P. M. Reutelingsperger, Tim G. A. M. Wolfs

**Affiliations:** 1Department of Pediatrics, Maastricht University Medical Center, 6202 AZ Maastricht, The Netherlands; r.gussenhoven@maastrichtuniversity.nl (R.G.); luise.klein@maastrichtuniversity.nl (L.K.); d.ophelders@maastrichtuniversity.nl (D.R.M.G.O.); b.kramer@maastrichtuniversity.nl (B.W.K.); 2School for Mental Health and Neuroscience (MHeNs), Maastricht University Medical Center, 6229 ER Maastricht, The Netherlands; 3School of Oncology and Developmental Biology (GROW), Maastricht University Medical Center, 6229 ER Maastricht, The Netherlands; denise.habets@maastrichtuniversity.nl; 4Department of Obstetrics and Gynecology, Maastricht University Medical Center, 6202 AZ Maastricht, The Netherlands; 5Institute for Transfusion Medicine, University Hospital Essen, University of Duisburg-Essen, 45147 Essen, Germany; bernd.giebel@uk-essen.de; 6Department of Biochemistry, Cardiovascular Research Institute Maastricht (CARIM), Maastricht University Medical Center, 6200 MD Maastricht, The Netherlands; l.schurgers@maastrichtuniversity.nl; 7Department of Biomedical Engineering (BMT), School for Cardiovascular Diseases (CARIM), Maastricht University Medical Center, 6229 ER Maastricht, The Netherlands

**Keywords:** Annexin A1/Formyl peptide receptor axis, blood-brain barrier, mesenchymal stem cell-derived extracellular vesicles, neonatal hypoxia-ischemia, therapy

## Abstract

Blood-brain barrier (BBB) disruption is associated with hypoxia-ischemia (HI) induced brain injury and life-long neurological pathologies. Treatment options are limited. Recently, we found that mesenchymal stem/stromal cell derived extracellular vesicles (MSC-EVs) protected the brain in ovine fetuses exposed to HI. We hypothesized that Annexin A1 (ANXA1), present in MSC-EVs, contributed to their therapeutic potential by targeting the ANXA1/Formyl peptide receptor (FPR), thereby preventing loss of the BBB integrity. Cerebral ANXA1 expression and leakage of albumin into the fetal ovine brain parenchyma after HI were analyzed by immunohistochemistry. For mechanistic insights, barrier integrity of primary fetal endothelial cells was assessed after oxygen-glucose deprivation (OGD) followed by treatment with MSC-EVs or human recombinant ANXA1 in the presence or absence of FPR inhibitors. Our study revealed that BBB integrity was compromised after HI which was improved by MSC-EVs containing ANXA1. Treatment with these MSC-EVs or ANXA1 improved BBB integrity after OGD, an effect abolished by FPR inhibitors. Furthermore, endogenous ANXA1 was depleted within 24 h after induction of HI in cerebovasculature and ependyma and upregulated 72 h after HI in microglia. Targeting ANXA1/FPR with ANXA1 in the immature brain has great potential in preventing BBB loss and concomitant brain injury following HI.

## 1. Introduction

Hypoxic ischemic encephalopathy (HIE) in new-borns is defined as a brain injury caused by insufficient blood flow and oxygen supply to the brain generally, due to placental insufficiency or umbilical cord occlusion in the perinatal period [1]. HIE is therefore after preterm birth the most important recognized perinatal cause of neonatal mortality [2]. Accordingly, HIE has detrimental effects on brain development with severe long-term disabilities, including cognitive impairments, mental retardation, epilepsy and cerebral palsy [1]. Treatment strategies are currently limited to hypothermia therapy which is only applicable in late preterm and term infants with HIE [3,4,5,6].

Understanding the underlying pathologic mechanisms of HIE is crucial for the development of new therapeutic interventions suited for preterm infants as well. Besides direct effects of hypoxia and reperfusion, and chronic neuroinflammation, the disruption of the blood-brain barrier (BBB) is increasingly recognized as an important cause of brain injury following neonatal HIE [5,7,8,9,10]. The BBB is mainly composed of endothelial cells (ECs) which form a barrier that communicates with both, the peripheral immune system and cells of the central nervous system, including pericytes, neurons, microglia and astrocytes, to maintain cerebral homeostasis [11,12]. There is evidence that the BBB is already functional early in development [13]. Following a hypoxic-ischemic (HI) insult excessive glutamate, due to energy failure is released and free radicals are produced damaging cerebral cells, including ECs of the BBB [1,11]. Necrotic and apoptotic cells activate microglia that release cytokines [1]. Microglia activation and oxidative stress contribute to changes in BBB permeability [14,15]. BBB permeability increases and perpetuates the neuroinflammatory response by enabling the infiltration of peripheral immune cells potentially aggravating the immune response and brain injury [16,17]. Clinical studies analyzing the BBB integrity demonstrate that global HI in neonates resulted in an increased cerebrospinal fluid (CSF)/serum albumin-ratio, an indicator for increased BBB permeability, and subsequent deregulation of cerebral homeostasis [7,8]. Moreover, more severe HIE clinical scores are associated with increased BBB permeability measurements, suggesting that brain injury is exacerbated, due to BBB disruption [7]. In line, experimental rodent and ovine models for HI demonstrated that HI adversely affected the BBB. More precisely, brain pathology was closely related to areas of compromised BBB [18], and a decrease in tight junction proteins was correlated with increased BBB permeability [9]. Altogether, the contribution of BBB injury in the pathophysiology of HIE indicates that strengthening the BBB integrity in the course of HI can attenuate brain injury. This idea mandates the understanding of the mechanisms regulating BBB integrity in order to exploit them as potential therapeutic targets. 

We have previously shown in a fetal ovine HIE model that mesenchymal stem/stromal cell derived extracellular vesicles (MSC-EVs) improved brain stem function and partially protected against hypomyelination. The therapeutic effects could not be attributed to the anti-inflammatory effects of MSC-EVs [19]. Considering the role of the BBB, we hypothesized that the protective effects of MSC-EVs against global HI induced brain injury involve the protection of the BBB. 

Recently, Solito et al. identified Annexin A1 (ANXA1) to be an essential endogenous regulator of BBB integrity in neurodegenerative diseases [20]. This group furthermore showed that pharmacological administration of human recombinant ANXA1 (hrANXA1) ameliorated BBB permeability [20]. Classically, ANXA1 is appreciated as an important downstream effector molecule of anti-inflammatory glucocorticoid effects [21]. It activates anti-inflammatory and pro-resolving mechanisms mainly as an agonist of formyl peptide receptors (FPRs), which are G protein-coupled receptors regulating host defense and inflammation [22,23]. In the adult brain, changes in ANXA1 expression are associated with neurodegenerative diseases related to BBB dysfunction, such as Alzheimer’s disease, Parkinson’s disease and multiple sclerosis [20,24,25,26,27]. In the fetal brain, ANXA1 is expressed by microglia and ECs that form the BBB [12], suggesting a role of ANXA1 in cerebral inflammation and BBB regulation during development. Since annexins are frequently found in the proteome of extracellular vesicles [28,29,30], we hypothesized that MSC-EVs protected BBB integrity after global HI in new-borns, at least in part, through ANXA1 that is present in MSC-EVs [28,29,30]. We tested this in a preterm fetal ovine model in which brain injury was induced by global HI by transient umbilical cord occlusion (UCO) for 25 min [16,17,19]. Furthermore, we studied mechanistic insights in a widely accepted BBB in vitro model in which trans-endothelial electrical resistance (TEER) was measured on fetal primary ECs as a marker for endothelial integrity [31].

## 2. Experimental Section

### 2.1. Study Approval and Experimental Design

All experimental procedures were performed in compliance with a protocol approved by the Animal Experiments Committee of Maastricht University Medical Center, NL (Mesenchymal stem cell-derived exosomes as a therapeutic intervention for hypoxia-ischemia in the preterm neonate, DEC 2012-064) and conducted in accordance with ARRIVE guidelines (https://www.nc3rs.org.uk/arrive-guidelines) and the Maastricht University guidelines on the Care and Use of the Laboratory Animals. The study design was described previously [19]. Briefly, individual fetuses (*n* = 37) of Texel pregnant ewes were randomly assigned to four different experimental groups: (1) Sham UCO and saline treatment (sham-SAL), (2) sham UCO and MSC-EV treatment (sham-MSC-EVs), (3) UCO and saline treatment (HI-SAL) and (4) UCO and MSC-EV treatment (HI-MSC-EVs). All fetuses were instrumented (IN) at 102 days (d) of gestational age (GA, term ~147 days of gestational age). An inflatable vascular occluder was inserted around the umbilical cord for induction of transient global HI. An umbilical vessel catheter was placed in the femoral artery and brachial vein for measuring blood pressure and for the administration of MSC-EVs, respectively. After a recovery period of four days, fetuses were subjected to 25 min of sham occlusion or UCO through inflation of the vascular occluder. At the time of UCO, ovine brain development is comparable with 30 weeks of gestation in humans which coincides with the high-risk window of brain injury in preterm infants [32]. Fetuses assigned for MSC-EVs treatment received two boluses of MSC-EVs from 2 × 10^7^ cell equivalents at 1 h and 4 days following (sham) UCO. Control animals received an equal volume of sterile 0.9% sodium chloride (SAL) at similar time points. Fetuses were sacrificed (END) after 1 day (*n* = 10), 3 days (*n* = 8) or 7 days (*n* = 19) after (sham) UCO (Figure 1). The investigators performing the (sham) UCO, tissue sampling and post-mortem analysis were blinded to treatment allocation. The MSC-EV groups on day 7 are relatively small, due to a dropout of 16% which was primarily restricted to the sham-MSC-EV group (i.e., three animals of the sham-MSC-EV group and 1 animal of the HI-MSC-EV) caused by a technical reason as reported previously [19].

### 2.2. Mesenchymal Stem/Stromal Cell-Extracellular Vesicles

After informed consent according to the Declaration of Helsinki, bone marrow-derived MSCs were raised from human donated bone marrow, as described previously [33]. The MSC characteristics were verified by flow cytometry and conventional MSC differentiation assays [19,33]. MSC-conditioned media was harvested as described previously and EVs were harvested from MSC-conditioned medium through a polyethylene glycol (PEG) method in which following PEG precipitation the volume of the conditioned media is reduced by a low-speed centrifugation (1500× *g*) after overnight incubation at 4 °C. Obtained pellets were resuspended and washed in 0.9% NaCl and following reprecipitation by ultracentrifugation solved in 1 mL aliquots, reflecting the MSC-EV yield of the supernatant of 4 × 10^7^ MSCs [33]. MSC-EVs were characterized by Nanoparticle Tracking Analysis (NTA) and by western blots for CD81 and tumor susceptibility gene 101 (Tgs101), as described previously [33,34]. For our study, MSC-EVs were further characterized by tunable resistive pulse sensing (TRPS) and additional western blot analyses.

### 2.3. MSC-EV Analysis Using Tunable Resistive Pulse Sensing (TRPS)

For in vitro experiments, amount of particles within the MSC-EVs were determined by TRPS using a qNano Gold with Izon Control Suite 3.2 Software and CPC100 calibration beads (Izon, Oxford, UK) by the Department of Medical Microbiology at Maastricht University. An NP150 nanopore (Izon, Oxford, UK) was coated using the Izon reagent kit for EV analysis according to the manufacturer’s instructions. To obtain a stable baseline current, samples were diluted 1:100 in Solution Q (Izon, Oxford, UK) and to prevent frequent pore obstruction, 10% (*v*/*v*) Solution G (Izon, Oxford, UK) was added. The NP150 nanopore was used at a stretch of 46.51 mm and a pressure of 6 mbar. Mean current for measuring was 128 nA with a voltage of 0.42 V. Recordings were stopped after 10 minutes with more than 400 blockades detected in both samples.

### 2.4. Western Blot of MSC-EVs

MSC-EVs (40 μL, 1.76 mg/mL protein concentration determined by Pierce™ BCA assay (Fisher Scientific, Landsmeer, The Netherlands)) were tested for ANXA1 by western blot analysis. Pure human platelet lysate (hPL), hPL centrifuged at 10,000× *g* for 10 min and centrifuged lysate with subsequent filtration through a 0.2 µM filter was used as a negative control to ensure that no platelet derived ANXA1 was present in the MSC-EVs as MSCs were expanded in MSC basal media supplemented with 10% hPL. Gel electrophoresis of samples was performed on a 10% SDS-polyacrylaminde gel. 

Separated samples, including 200 ng hrANXA1 [35] and a broad protein marker (Biorad, Hercules, CA, USA), were transferred to a nitrocellulose membrane and probed with a 1:1000 dilution of anti-ANXA1 which was a kind gift from Prof. Mauro Perretti followed by a goat anti-rabbit-IgG-alkaline phosphatase (Sigma-Aldrich, St. Louis, MO, USA). For visualization the membrane was washed in a reaction buffer (0.1 M Tris-HCl, 0.1 M NaCl, 5 mM MgCl_2_) followed by incubation in substrate solution containing Nitrotetrazolium Blue chloride (Sigma-Aldrich, St. Louis, MO, USA) and 5-Bromo-4-chloro-3-indolyl phosphate disodium salt (Sigma-Aldrich, St. Louis, MO, USA). The reaction was stopped by 0.5 M EDTA, and scans were taken by an Epson Perfection V300 Photo scanner (Appendix A).

This western blot procedure was confirmed and extended. For this experiment, a dilution curve of 1, 3, 10 and 30 ng hrANXA1 [35] and a Dual color marker (Biorad, Hercules, CA, USA) were transferred to a nitrocellulose membrane and probed with a 1:2500 dilution of anti-ANXA1 (kind gift from Prof. Mauro Perretti) followed by a goat anti-mouse-IgG-HPR (Dako, Santa Clara, CA, USA). Samples were detected by Pierce™ ECL Western Blotting according to the manufacturer’s protocol (Fisher Scientific, Landsmeer, The Netherlands). Pictures were taken by an Odyssey scanner at different exposure times (50 s and 1500 s) (Li-cor, Bad-Homburg, Germany).

### 2.5. Immunohistochemistry and Analysis

Immediately after sacrifice, fetal brains were removed from skulls and weighed. The right hemisphere was submersion fixated in ice-cold 4% paraformaldehyde for three months. After fixation, a predefined region containing the lateral ventricles, periventricular white matter and basal ganglia was embedded in paraffin and serial coronal sections (4 µm) were cut with a Leica RM2235 microtome (Leica Microsystems B.V., Amsterdam, The Netherlands). Coronal sections were stained for albumin as a marker for BBB leakage, ionized calcium binding adaptor molecule 1 (IBA1) as a general microglia marker and ANXA1. Endogenous peroxidase activity was quenched via incubation with 0.3% hydrogen peroxide dissolved in Tris-Buffered Saline (TBS). Antigen retrieval involved boiling tissues in a sodium citrate buffer (pH 6.0) using a microwave oven. Next, sections were incubated overnight with the primary polyclonal rabbit anti-ANXA1 (Abcam; Cambridge, UK, 1:100), anti-albumin (Accurate Chemical, Westbury, NY, USA; 1:2000), anti-IBA1 (Wako chemicals, Neuss, Germany; 1:1000) antibody at 4 °C, followed by incubation with a secondary polyclonal swine anti-rabbit biotin (Dako, Santa Clara, CA, USA; 1:200). The antibody specific staining was enhanced with a Vectastain ABC peroxidase elite kit (Vector Laboratories, Burlingame, CA, USA) followed by a 3,3′-diaminobenzidine (DAB) staining. Nuclei were counterstained with Mayer’s haematoxylin. 

Analysis of immunohistochemical stainings was done after taking digital images using a Leica DM2000 microscope with Leica Qwin Pro V3.4.0. Software (Leica Microsystems, Mannheim, Germany). Images of ANXA1 and IBA1 were taken at a magnification of 100×. Region of interest comprised the blood vessels, ependymal lining cells and white matter, including microglial cells stained with IBA1. Assessment of ANXA1 IR in microglial cells was determined based on cellular phenotype and staining of adjacent sections with IBA1 co-localizing with ANXA1 IR (Figure 2a,b). For the quantification of the intensity of ANXA1 IR we used a scoring system (1-3) to evaluate the immunoreactivity intensity of ANXA1 whereby score 1 comprised minor, 2 moderate and 3 intense immunoreactivity (IR) (Figure 2c). Scoring was complemented by analysis of area fractions, expressed as the percentage of positive staining relative to the total area using a standard threshold intensity, determined with Leica Qwin Pro V 3.5.1. Software (Leica, Rijswijk, The Netherlands). Same area fraction measurements were used for IBA1 IR assessment as described previously [16]. Moreover, the thickness of the ANXA1 positive stained periventricular area was measured with ImageJ Software version 1.48. 

Analysis of albumin staining was done on 10 images (200× magnification) of similar sized blood vessels per animal. To evaluate the integrity of the BBB, albumin extravasation was scored with a (−) if no albumin was present in the cerebral parenchyma and a (+) if positive albumin staining was present in the surrounding cerebral tissue of the blood vessel (see exemplary Figure 3b). These results are displayed as a percentage of albumin extravasation indicating leaky blood vessels. 

### 2.6. Endothelial Cell Isolation and Culture

Endothelial cells were a generous gift of Dr. Nynke van den Hoogen, University of Maastricht, Department of Mental Health and Neuroscience. Cells were isolated from brains of Sprague-Dawley rat pups sacrificed at day P3 by cervical dislocation accordingly to Bernas et al. 2010 [36]. The brain developmental stage of rodents on postnatal day 3 (P3) is comparable to preterm human infants [37]. Brains were dissected from the skull and meninges and large vessels were removed before trituration of the tissue by passing the fragments through decreasing pipet tips. Large fragments were filtered out by passing cell suspension through a 500 µM strainer (pluriSelect, Leipzig, Germany). Cells in the flow-through were collected on a 30 µM strainer (pluriSelect, Leipzig, Germany) and subsequently centrifuged at 51× *g* for 10 min. The pellet was resuspended in DMEM-F12-glutamax (Fisher Scientific, Landsmeer, NL) supplemented with 10% heat inactivated fetal bovine serum (FBS) (Sigma-Aldrich, St. Louis, MO, USA), 1% antibiotic-antimycotic solution (Sigma-Aldrich, St. Louis, MO, USA), 50 µg/mL EC growth supplement (BD Biosciences, San Jose, CA, USA), 1 mg/mL heparin (Biochrom GmbH, Berlin, Germany) and hydrocortisone 500 nM (Stemcell Technologies, Cologne, Germany) and transferred into a T25 flask pre-coated with type-I-collagen (Corning Life Science, Oneonta, NY, USA). Culture expansion was allowed for approximately one month to achieve highly confluent cerebrovascular ECs showing minimal contamination by pericytes (<5%). Characterization of the ECs to assess the purity of the cell population was performed by immunocytochemistry. Cells were grown on glass slides and stained for von Willebrand Factor (vWF) (Dako, Santa Clara, CA, USA), zona-occludens 1 (ZO-1) (Fisher Scientific, Landsmeer, The Netherlands) as EC markers and α-smooth muscle actin (α-SMA) as marker for pericytes (Sigma-Aldrich, St. Louis, MO, USA). Cells were fixated by incubation in 4% paraformaldehyde followed by blocking with either 3% bovine serum albumin (BSA) or 10% normal goat serum (NGS) in phosphate buffered saline (PBS). Cells were then incubated overnight with the primary antibody (ZO-1 1:50, α-SMA and vWF 1:200) at 4 °C, followed by incubation with the appropriate alexa-fluor labelled secondary antibody (1:200). Nuclei were stained with DAPI and coverslips were mounted using a fluorescent mounting medium (Dako, Santa Clara, CA, USA). 

### 2.7. Trans-Endothelial Electrical Resistance (TEER) 

A cellular monolayer of ECs was cultured on semipermeable filter inserts (transwell, 12 wells) (Corning Life Science, Oneonta, NY, USA). TEER was measured as an established quantitative readout for barrier integrity [31] using an Epithelial Voltohmmeter (EVOM2, World Precision Instruments, Sarasota, FL, USA) with two chopstick electrodes, each containing a silver-silver chloride pellet for measuring voltage and a silver pellet for passing current. Measurements of the resistance in ohm (Ω) across the cell layer were made on the semipermeable membrane by placing one electrode in the upper compartment and the other electrode in the lower compartment. Measurements were performed in duplo per insert and consistently conducted for several days, 30 min after culture media was changed and the temperature was kept at 37 °C before and between all measurements. Once values plateaued, the membrane reached confluency and further experiments could be performed (baseline measurement). When ECs reached confluency in the transwells, cells were randomly assigned to oxygen-glucose deprivation (OGD) or normoxia conditions and baseline TEER was measured. Normoxia controls were left in normal culture conditions. OGD was performed by changing the culture media with DMEM without glucose and glutamine (Life Technologies, Carlsbad, CA, USA) and exposing ECs to 0% oxygen in a hypoxic chamber at 37 °C for four hours. After 4 h of normoxia/OGD, medium was changed to culture media and TEER was measured followed by one or a combination of the treatments at the following concentrations: 1.1 × 10^9^ MSC-EVs were given per well based on our TRPS analysis, hrANXA1 [31] (3 µM), FPR1/2 receptor blockers WRW4 (10 µM, TOCRIS Bio-techne Ltd., Abingdon, UK) and cyclosporine H (1 µM, Sigma-Aldrich, St. Louis, MO, USA). Subsequently, TEER was measured in all groups at 0, 3, 6, 12 and 24 h after normoxia/OGD. This setup resulted in following treatment groups: (1) No treatment, (2) MSC-EVs, (3) hrANXA1, (4) WRW4, (5) Cyclosporine H, (6) WRW4 + MSC-EV, (7) cyclosporine H + MSC-EVs, (8) WRW4 + hrANXA1, (9) cyclosporine H + hrANXA1. Cell culture experiments were repeated in duplo to test for reproducibility (*n* = 4). 

### 2.8. Statistical Analysis 

Regarding immunohistochemistry results, all values are shown as mean with 95% confidence interval (CI) or standard deviations (SD). Comparison between different experimental groups was performed with one- or two-way analysis of variance (ANOVA) and appropriate post-hoc testing. In case data were positively skewed, log-transformation was applied to obtain normal distributed data or non-parametric testing was performed.

Data from TEER measurements were obtained from two independent experiments each run in *n* = 2 per treatment group making a total of *n* = 4 per group. The absolute TEER values are reported as resistance across the EC layer on the semipermeable membrane in Ohm (Ω). These values are averaged and presented as mean absolute TEER measurements (Ω) with 95% CI and tested with a two-way analysis of variance (ANOVA) and appropriate post-hoc testing for significance.

Statistical analysis was performed with IBM SPSS Statistics Version 22.0 (IBM Corp., Armonk, NY, USA) graphical design was performed using GraphPad Prism 5 (GraphPad Prism, La Jolla, CA, USA). Exact p-values are reported and statistical significance was accepted at *p* < 0.05. 

## 3. Results

### 3.1. MSC-EVs Tended to Prevent Albumin Leakage in the Fetal Ovine Brain following Global Hypoxia-Ischemia

Global HI induced a 34–38% increase in albumin leakage compared to control animals (mean of 22% vs. 56% in sham-SAL vs. 7d HI-SAL *p* = 0.054 and mean of 22% vs. 60% of 3d HI-SAL *p* = 0.088) into the brain parenchyma (Figure 3a). At seven days, in four out of seven (56%) animals increased albumin leakage was found following HI compared to the mean of the sham-SAL group (sham-SAL vs. 7d HI-SAL, *p* = 0.054). No difference in albumin leakage was found at one day following HI (data not shown). MSC-EV treatment prevented albumin leakage on day 7 after reperfusion compared to the untreated control animals following HI (HI-SAL vs. 7d HI-MSC-EVs *p* = 0.0501). MSC-EVs did not induce albumin leakage in sham conditions (sham-SAL vs sham-MSC-EVs *p* = 0.613) (Figure 3a).

### 3.2. MSC-EVs and ANXA1 Restored Endothelial Resistance/Barrier Integrity following Oxygen-Glucose Deprivation in FPR Dependent Manner

To study the potential mechanism underlying the protective effects of MSC-EVs observed in vivo, a model of primary fetal ECs isolated from rat brains at postnatal day 3 (P3) was used. Morphology of ECs was typically cobblestone-like (Figure 4a). Purity of EC culture was determined, four weeks after starting of the cell culture by immunocytochemical analysis of the vWF (Figure 4b), ZO-1 (Figure 4c) and pericyte marker α-SMA (Figure 4d). Baseline TEER values were approximately 150 Ω per trans-well insert before experiments were initiated. After four hours of OGD, TEER values significantly decreased in each treatment group. Subsequently, MSC-EVs steadily increased TEER and values plateaued from 12 h onwards at 122 Ω (no treatment vs MSC-EVs 24 h, *p* = 0.022). These improved TEER values upon MSV-EV treatment were not detected when the FPR1 and FPR2 receptors were blocked by cyclosporine H and WRW4 (Figure 4e,f). As ANXA1 is an appreciated agonist of FPRs and known to strengthen BBB integrity in adult neuropathologies, we investigated whether ANXA1 is present in MSC-EVs and confirmed its presence by a western blot (Figure 4g). Taking pictures at low exposure times (10–50 s), 10 and 30 ng hrANXA1 (37 kDa) was already detectable. After increasing the exposure time (1500 s), a fragment of 37 kDa was detected in the MSC-EV sample whereas still no band was visible in the 1 and 3 ng hrANXA1 lanes or negative controls. Since the MSC-EV fraction used for the blot (40 µL) contains more than 3 ng and less than 10 ng hrANXA1, 1 mL of MSC-EVs isolated from 4 × 10^7^ cells would contain (75–250 ng ANXA1/mL aliquot MSC-EVs. In line, a physiological relevant amount of ANXA1 in MSCs was found within the same ng range [30].

Subsequently, we tested whether BBB integrity in our in vitro model was improved after OGD in response to hrANXA1. Similar to MSC-EV treatment, hrANXA1 significantly improved TEER values after OGD (no treatment vs MSC-EVs 24 h, *p* = 0.017), which was prevented in the presence of FPR inhibitors (Figure 4h,i).

### 3.3. ANXA1 is Widely Expressed in the Preterm Ovine Brain and the Expression Decreases Acutely after Global Hypoxia-Ischemia

We investigated the temporary dynamics of ANXA1 expression within cerebral blood vessels, ependymal lining and microglia in vivo after HI. We found that one day after global HI, ANXA1 IR decreased significantly in blood vessels (1d sham-SAL vs. 1d HI-SAL *p* = 0.032) and ependymal lining cells (1d sham-SAL vs. 1d HI-SAL *p* = 0.007) compared to controls, whereas at day 3 and day 7 seven ANXA1 expression were normalized (3d sham-SAL vs. 3d HI-SAL *p* = 0.880 and *p* = 0.975; 7d sham-SAL vs. 7d HI-SAL *p* = 0.100 and *p* = 0.894) (Figure 5a,b). Systemic administration of MSC-EVs did not change ANXA1 expression seven days after (sham) UCO in cerebral vasculature, ependyma and microglia (7d HI-SAL vs. 7d HI-MSC-EVs *p* = 0.794, 7d HI-SAL vs. 7d HI-MSC-EVs *p* = 0.603, 7d HI-SAL vs. 7d HI-MSC-EVs *p* = 0.999) (Appendix A). 

Since ANXA1 is expressed by microglia, we assessed IBA1 IR and ANXA1 IR in adjacent sections in the same regions of interest as illustrated in Figure 2a,b. In these white matter regions microglia were abundantly present and we found that at three days after global HI ANXA1 expression was significantly increased compared to time-matched controls (3d sham-SAL vs. 3d HI-SAL *p* = 0.047) (Figure 5c). Moreover, analysis of area fraction of IBA-1 IR showed that increased microglial activity was found at three and seven days following HI (data not shown) which is in line with previous studies [16,19,38,39]. 

## 4. Discussion

We show in our translational fetal sheep model of global HI that the BBB integrity is compromised, as indicated by an increased albumin leakage into the brain parenchyma. This is in line with clinical reports and studies with fetal animals in which loss of BBB integrity occurred after HI [7,9,10,18]. BBB dysfunction is one of the key mediators in adult neurological disorders, including multiple sclerosis (MS), stroke, Alzheimer’s and Parkinson’s disease, since it facilitates immune cell infiltration into the central nervous system (CNS), which acts as drivers of cerebral inflammation and subsequent brain injury [24,25,26,27,40]. In the fetal situation, HI results in a multifactorial cascade of detrimental events, of which BBB dysfunction similar to adult neuropathologies is identified as one of the contributors to cerebral injury leading to life-long cognitive and sensory motoric disabilities [7,8]. In addition to direct BBB dysfunction, multiple phases of energy failure following global HI lead to cell death and release of excitotoxic molecules, eliciting a strong pro-inflammatory response by microglia that secrete cytokines and reactive oxygen species that exacerbate existing BBB damage [1,14,15,25,41,42,43]. Moreover, systemic inflammation, induced by a global HI insult, can increase the BBB permeability, allowing inflammatory mediators, such as cytokines and inflammatory cells to enter the parenchyma, thereby contributing to disease progression. Evidence for such second hit in our global HI model was previously shown by recruitment of immune cells into the circulation within 24 h following global HI with a subsequent marked cerebral influx of neutrophils and T-cells [16]. The influx of peripheral immune cells after cerebral ischemia is considered to further aggravate acute inflammation of the brain that was initiated by immediate cell death and microglial activation [42,43,44]. This emphasizes that disturbance of BBB integrity by HI and peripheral and cerebral inflammation act synergistically, leading to a self-enhancing loop of inflammation/brain damage which needs to be contained. Hence, targeting molecular pathways regulating BBB integrity appears a logical strategy to combat brain injury following HI.

We have previously shown in our fetal sheep model of global HI that pharmacological administration of MSC-EVs resulted in partial protection against hypomyelination. These therapeutic effects could not be explained by the anti-inflammatory effects of the MSC-EVs [19]. Therefore, we focused on an alternative explanation for the pharmacological effects of MSC-EVs, being restoring the injured BBB by targeting the ANXA1/FPR-axis following HI. In the current study, we report that MSC-EVs protect the BBB that was compromised by global HI in our fetal sheep model. We conclude from our previous [19] and current findings that the protective effects of MSC-EVs arise, at least in part, from their protective actions on the BBB. Interestingly, recent studies of adult traumatic brain injury and stroke models showed that stem cell therapy diminished trauma and HI-induced BBB leakage [44,45]. Although our combined findings indicate that ANXA1-driven inhibition of cerebral inflammation is not a plausible explanation for the pharmacological effects of MSC-EVs, we cannot rule out the possibility that microglia are part of the underlying working mechanism. More precisely, Solito et al found that ANXA1 plays an important role in controlling non-inflammatory phagocytosis of apoptotic cells and promoting resolution of inflammation in models of Alzheimer’s disease [25]. An annexin-driven switch into M2 microglia that typically express such characteristics is a realistic scenario that warrants further investigation in a follow up study where neuroprotective effects of ANXA1 will be directly tested in vivo. 

The most important finding of this study was that MSC-EV mediated protection of BBB integrity (after OGD) was dependent on FPR signaling and that MSC-EVs contain the potent FPR agonist ANXA1. ANXA1 is a multifunctional molecule originally identified as an anti-inflammatory/pro-resolving mediator and more recently also as a crucial regulator of BBB integrity [20,21]. In particular, ANXA1 KO mice exhibit increased BBB permeability compared to WT mice, due to disrupted inter-EC tight junctions [20]. ANXA1 regulates the BBB by binding to the FPR2 receptor that in turn inhibits the activity of RhoA, subsequently promoting cytoskeletal stability and enhancing tight junction formation [20,46,47]. Blocking of FPR signaling by cyclosporine H or WRW4 leads to activation of RhoA that might induce a leaky BBB by destabilizing the β-actin cytoskeleton. RhoA activity, inhibited by ANXA1, was also identified to induce BBB permeability in AD supporting the role of ANXA1 to regulate BBB integrity by this pathway [48]. Another study proposed FPR/ANXA1 interactions in improving BBB integrity, in a model studying sexual dimorphism in systemic inflammation [49]. Exclusively in young female mice ANXA1 signaling, downstream of estrogen, prevented inflammation-induced BBB tight junction breakdown and limited lymphocyte extravasation into the brain parenchyma [49]. Similar sexual dimorphism in susceptibility of HI induced preterm brain injury is also observed in the clinic and experimental models which found females to be better protected from premature brain damage and sequelae compared to males [50,51,52].

In accordance with previous reports, we found ANXA1 to be expressed by fetal microglia, BBB endothelial and ependymal cells [12,53,54]. The mechanism by which endogenous ANXA1 regulates BBB integrity in brain microvascular ECs is considered to occur by binding to β-actin, thereby promoting cytoskeleton formation, which serves as an anchor linking β-actin to the plasma membrane, facilitating the establishment of tight junctions between ECs [20]. 

Since ependymal cells, which form the basis of the brain-CSF interface, fulfil a similar barrier function during development as ECs of the BBB [13,55,56] it is likely that ANXA1 exerts the same regulatory function in these cells. Neuro-ependymal cells of the fetus are connected by strap junctions that restrict the influx of larger molecules from the CSF into the brain [13,55,56,57] and disruption might cause uncontrolled passage of larger molecules between the brain and the CSF. This tempts us to speculate that loss of ANXA1 within the first 24 h after HI might lead to a weakened brain-CSF barrier function. For future research, determination of albumin in CSF immediately after HI insult might give insights into the function of ANXA1 in the ependymal lining. We aimed to measure albumin in CSF, but unfortunately, this analysis was hampered by blood contamination during sampling of CSF in a number of samples.

An important observation in this study is the acute, transient decrease of endogenous ANXA1 expression of the cerebral vasculature following global HI, indicating that an important endogenous protection of the BBB is abrogated, which might destabilize BBB integrity promoting the vicious cycle of inflammation/brain damage. The clinical consequences of ANXA1 loss are supported by earlier findings, showing that depletion of ANXA1 was also detected in cerebrovascular capillaries and serum of MS and AD patients and respective experimental models in which a disrupted BBB is a major requirement for the onset of disease [20,48]. 

Besides acute ANXA1 loss in cerebrovasculature and ependymal cells, we found an increase in ANXA1 expression of microglia three days after HI compared to time-matched control animals. The increase of endogenous ANXA1 in microglia after the acute phase might be part of the controlled resolution of inflammation. ANXA1 facilitates phagocytosis of apoptotic cells and debris by microglia without eliciting a pro-inflammatory response [25,43,47,58]. Furthermore, ANXA1 skews macrophages from a pro-inflammatory towards an anti-inflammatory phenotype [59,60,61]. The specific and acute loss of endogenous ANXA1 following HI in cells lining the BBB suggests the idea that the BBB is the primary site affected by HI, highlighting it as the most obvious therapeutic target in HI.

A limitation of our in vitro study is that we cannot completely rule out that primary cells might lose characteristics during culturing. However, there is evidence that cerebral ECs maintain specific morphology in culture which is especially the case with the endothelium from embryonal origin [62]. In particular, fetal ECs did not show signs of cellular aging; cells retained ultrastructural characteristics and cells did not change lectin binding pattern [62]. Furthermore, there are several differences between adult and fetal brain ECs i.e., adult and fetal ECs differ in growth rate and expression of endothelial markers [63]. Moreover, functional differences have been reported which are important to be mentioned, such as different reaction to tumor-conditioned media [63] and neonatal ECs display a lower barrier integrity compared to adult cells measured by TEER [64]. 

Given the relatively small animal numbers per group, which is an inherent limitation to large animal models, we report actual P values and tend to interpret *p* values between 0.05 and 0.1 as biologically relevant. This assumption will decrease the chance of a false negative finding, but increases the chance that one of these differences is a false positive result.

Together, our data in this proof of concept study support the notion that the ANXA1/FPR axis is a therapeutic target to treat fetuses exposed to HI of the brain and that hrANXA1 is a potential therapeutic agent especially with regard to BBB function. Our current understanding of the pathologic course of HI is described by a primary phase during/after HI that is characterized by a reduction in cerebral blood flow and subsequent oxygen/nutrient shortage resulting in severe tissue damage. The primary phase is followed by a latent/recovery phase (~6 h after HI) in which cerebral blood flow is restored leading to a secondary energy failure phase 6–72 h after the initial insult [1,4,6]. The latent phase is believed to be the optimal window of therapeutic interventions [1,4,6]. Considering that ANXA1 depletion occurs within the first 72 h upon ischemia in cerebrovasculature and ependyma, supplementation of exogenous ANXA1 should be conducted within the pharmacological window of opportunity of at most three days after the HI insult, but preferably earlier. 

## Figures and Tables

**Figure 1 jcm-08-00137-f001:**
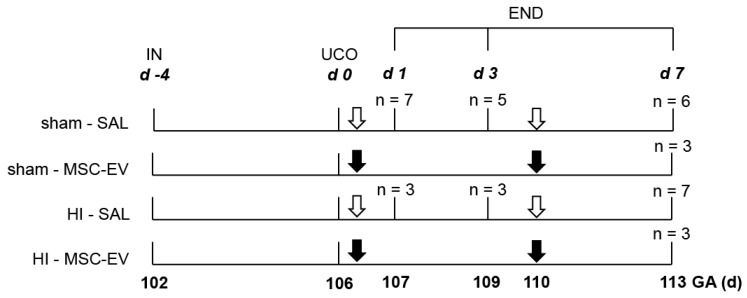
Experimental design. Fetuses were instrumented at GA 102 (d-4). After four days of recovery, fetuses were subjected to 25 min of umbilical cord occlusion (UCO) or sham occlusion (0 d). One hour and 4 days (110 d GA) after (sham) UCO, fetuses received either intravenous MSC-EVs (2.0 × 10^7^; cell equivalents; black arrows) or an equivalent volume of saline 0.9% (SAL; white arrows). After a 1 d, 3 d (sham-SAL and HI-SAL groups only) and 7 d (all groups) reperfusion period, at 107 d, 109 d and 113 d GA respectively, animals were sacrificed and brain tissue was collected. END—end of experiment; GA—gestational age; HI—hypoxia-ischemia; IN—instrumentation; MSC-EV—mesenchymal stem cell-derived extracellular vesicle, SAL—saline.

**Figure 2 jcm-08-00137-f002:**
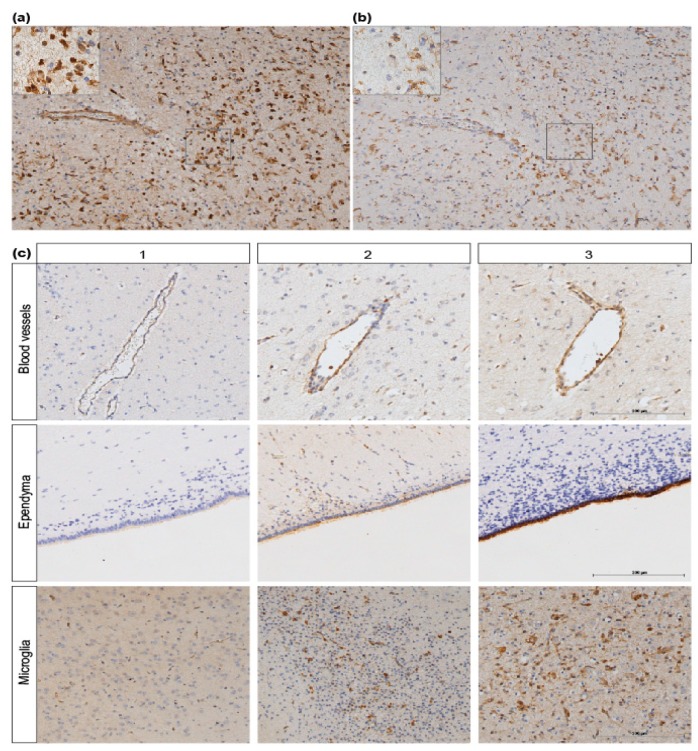
ANXA1 scoring system. (**a**) ANXA1 is expressed within the white matter and (**b**) co-localizes with IBA1 IR in adjacent sections (magnification 100×, scale bar 200 µm). (**c**) ANXA1 IR scoring system (1, 2, 3) of blood vessels, ependymal tissue and microglia (magnification 100×, scale bar 200 µm).

**Figure 3 jcm-08-00137-f003:**
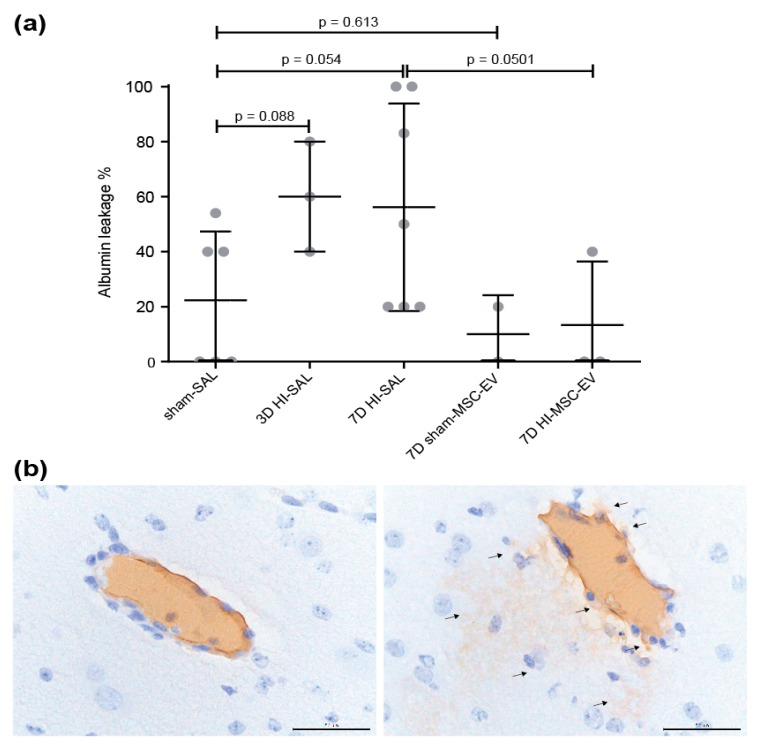
Albumin leakage into brain parenchyma. (**a**) Percentage of albumin leakage inside the brain parenchyma is depicted. (**b**) Immunohistochemical distribution of endogenous albumin. Representative vessels of an HI-MSC-EV animal showing albumin present inside the vessel (left) and albumin extravasation after HI (right) indicated by arrows. 400× magnification, scale bar 50 μm.

**Figure 4 jcm-08-00137-f004:**
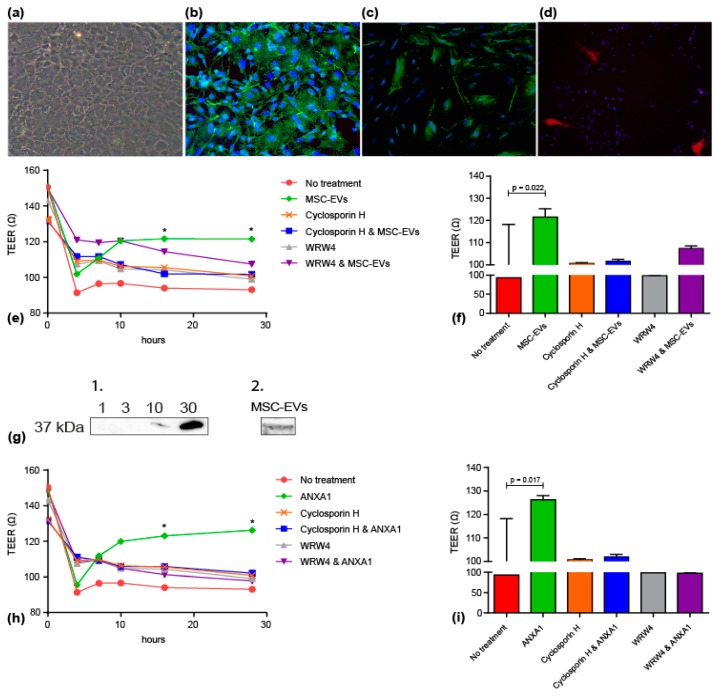
MSC-EVs and ANXA1 prevent loss of BBB integrity in vitro after OGD via FPRs. Characterization of primary fetal rat endothelial cell culture by (**a**) bright-field microscopy (100× magnification) and (**b**) immunocytochemistry for vWF (200× magnification) (**c**) immunocytochemistry for ZO-1 (200× magnification) (**d**) immunocytochemistry for α-SMA (40× magnification). (**e**) Baseline TEER measurements were taken before initiation of OGD (0 h). Four hours after OGD, fetal rat ECs were treated with MSC-EVs and/or FPR inhibitors and followed up for 3, 6, 12 and 24 h (*n* = 4). (**f**) TEER endpoint measurement 24 h after OGD and MSC-EV treatment (28 h absolute time) in the presence or absence of FPR inhibitors (*n* = 4); * = *p* < 0.05. (**g**) Western blot analyses to detect ANXA1 in MSC-EVs. g1. After 50 s of exposure time, a weak and intense fragment was detected in the 10 and 30 ng hrANXA1 lanes respectively whereas no signal was detected when MSC-EVs (not shown) or 1 and 3 ng hrANXA1 were loaded per lane. g2. After 1500 s of exposure time, endogenous ANXA1 (37 kDa) was detected in lysate of MSC-EVs, but not in MSC culture medium controls (pure hPL, pure hPL + 10,000× *g* for 10 min, pure hP +10,000× *g* for 10 min + 0.2 µM filtered) which were used as negative controls (not shown). (**h**) Baseline TEER measurements were taken before initiation of OGD (0 h). Four hours after OGD, fetal rat ECs were treated with hrANXA1 and/or FPR inhibitors and followed up for 3, 6, 12 and 24 h (*n* = 4). (**i**) TEER endpoint measurement, 24 h after OGD and hrANXA1 treatment (28 h absolute time) in the presence or absence of FPR inhibitors (*n* = 4); * =*p* < 0.05.

**Figure 5 jcm-08-00137-f005:**
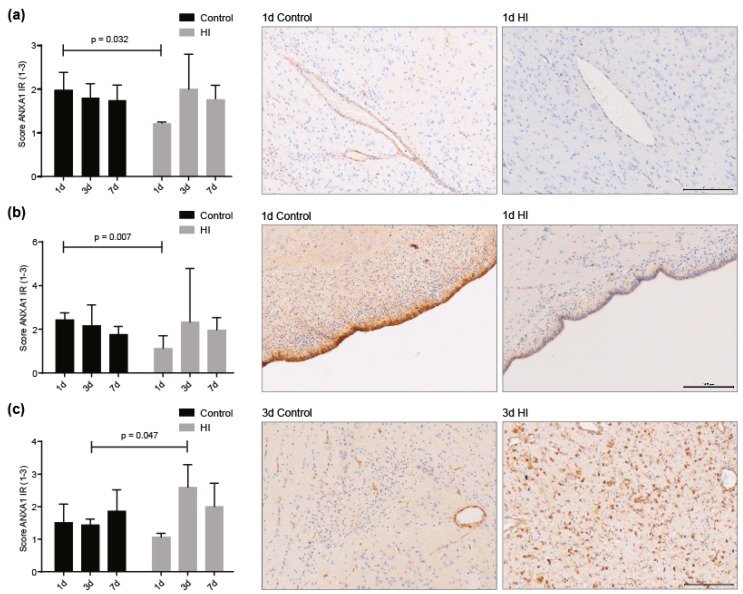
Temporal expression of ANXA1 on 1, 3 and 7 days after HI in cerebrovasculature, ependymal lining and microglia. (**a**) ANXA1 expression in cerebrovasculature and representative picture of ANXA1 loss in vasculature one day after HI compared to control. (**b**) Endogenous ANXA1 over time in ependymal lining and representative picture of ANXA1 loss in ependymal lining one day after HI compared to control. (**c**) Endogenous ANXA1 expression in microglia over time and representative pictures of ANXA1 increase in microglia three days after HI compared to control. 100× magnification, scale bar 200 μm.

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
