# Peer review of "Annexin A1 as Neuroprotective Determinant for Blood-Brain Barrier Integrity in Neonatal Hypoxic-Ischemic Encephalopathy"

_jcm, 2019, doi:10.3390/jcm8020137_

Round 1

Reviewer 1 Report

In their paper, Gussenhoven et al. provide new data that further complement the novel world of therapeutic interventions using extracellular vesicles. They here aim to show that the protein ANXA1, present in stem cell derived vesicles is directly responsible for improving BBB failure and cerebral inflammation in a model of perinatal hypoxic ischemic encephalopathy. As these events are central to many – if not all- neuropathologies, such therapeutic intervention may be extrapolated in future to more diseases that affect the CNS.

Nevertheless, although the underlying story is exciting and new, the manuscript suffers from some important gaps and uncertainties. Filling these gaps and clarifying some issues will gravely increase our understanding of what happens in the brain and expand its impact in the scientific world of barrier research (and beyond). Please find my detailed comments below.

Introduction/Discussion
“Hypoxic ischemic encephalopathy (HIE) in new-borns is defined as brain injury caused by insufficient blood flow and oxygen supply to the brain in the perinatal period” (line 46)

    Regardless its cause? This is not so much important as such, especially not to kick-off the manuscript, but I do run into problems when BBB dysfunction comes into play. What I’m hinting to is the following: is anything known on whether there is an intrinsic defect in BBB development leading to flawed perfusion and hence, HI? Or is BBB dysfunction consequent to HI in these infants? Are there classifications in the cause of HIE?

    By choosing the UCO models, authors here use a model for ischemia preceding BBB dysfunction. In this model, I also wonder whether the effects on the cerebral barriers are directly mediated by cerebral ischemic events? By occluding the umbilical cord, all foetal organs will suffer from ischemia. Could this induce a systemic inflammatory response that impacts on the BBB? Please comment. (Partly answered in discussion)

    Also the statement that “experimental rodent and ovine models for HI demonstrated that HI adversely affected the developing BBB” (line 68) is dubious considering we are talking about the perinatal period. It has been pointed out that the BBB is already functional at this stage. Please see the work of Norman Saunders on this matter. (E.g. Saunders et al Front Pharmacol 2012, Barrier mechanisms in the developing brain.) (same remark for the blood-CSF-barrier – see further comments).

Materials and Methods
“Immunohistochemistry and analysis: After fixation,….”

    Could the authors please first describe how brains are harvested and treated? (Everything that precedes fixation of the brain tissue?)

“The brain developmental stage of rodents on postnatal day 3 (P3) is comparable to preterm human infants”.
    And how does this compare to the ovine foetuses used in the in vivo part of the research? Also, how do these foetuses compare to human infants then?

    I am not contesting that rodent P3 may be comparable to the human preterm infant stage but I do seriously doubt whether this is still of importance once cells are cultured. Although primary cells do provide a good means to study some BBB characteristics in vitro, they also loose many of the characteristics for which reason I do not believe that we can still pinpoint a developmental stage on cultured BBB cells.

Results
Section 3.1 and Figure 2: I encourage the authors to delineate the endothelial boundary of the cerebral vessel in the examples. It is very though to decide whether there is extravasation of albumin or rather a morphology switch/irregularity of the endothelial monolayer (which is often observed in disease context). Therefore it is not convincing. Please also provide an example of the HI + EV condition. Given that foetuses were sacrificed at D1,
D3 and D7 and only results for D7 are shown, I take it that at D1 and D3 EV treatment had no success? Can authors please also provide this information?

Can FPR inhibitors be administered in vivo? What is their impact on barrier leakage? (and on microglial activation – see further)?

Section 3.2 and Figure 3: Combining Figure 3E and 3H does not provide sufficient evidence that ANXA1 is responsible for the protective effect of EVs. Also, in all honesty, figure 3g is not very convincing in indicating that EVs contain ANXA1. I have seen much more clear examples in a previously published paper in which the authors were involved.

Either way, in my opinion, a much better control (for both this section and for section 3.1) would be to deplete the EVs form ANXA1 (or harvest EVs from ANXA1-depleted cells, e.g. using Crispr or siRNA technology) and to evaluate their (lack of) therapeutic effect in terms of albumin leakage in vivo and TEER in vitro.

Section 3.3 and Figure 4: Panel C does not show actual results but rather the scoring system. I would place this in the M&M section as it is quite confusing here. What would be more appropriate is to show the reader examples of ANXA1 in Sham and HI brains (at different days in different cell types).

It is strange to me that, albeit authors present data on the ependymal cell layer and discuss it a bit, they base the remainder of their text entirely on the BBB and cerebral microvessels. They however fail to introduce and investigate possible consequences of ANXA1 loss and EV treatment on the blood-CSF barrier, a high important second barrier that protects the cerebral tissue. What is the rationale of presenting data on the ependymal cell layer? Although ependymal cells are not the basis of the B-CSF-B (this is wrongly stated in line 373, the choroid plexus epithelium is the sole barrier-forming cell type here), the images in Figure 4C are not detailed enough to distinguish ependymal cells from choroid plexus epithelium. Is anything known on this subject? Do authors detect leakage of the blood-CSF-barrier in their foetuses (e.g. presence of albumin in CSF would give a good indication as albumin seems not to spread too far out of the vessels)? This data should be added to section 3.1 if relevant.

Also the matter of microglia should be handled in more detail. Are microglia activated in the UCO animals? Does silencing microglia (e.g. using minocyclin) protect against barrier leakage in UCO animals? Can ANXA1-containing EVs obliterate microglia activation? (and ANXA1-depeleted EVs not?)

Author Response

Response to reviewer #1

We thank the reviewer for her/his interest in our work and appreciation of our new insights into the role of the BBB around HIE. Please find enclosed a point to point response to the comments of the first reviewer.

Reviewer 2 Report

The experiments reported in this manuscript are of scientific interest
However, the reporting of these experiments is far from being optimal and I have major concerns on several points:
1-in vivo experiment
(1) In vivo experimental design: even if times and treatments are well explained, the difference in the number of animals per group is too much variable and this point is not clarified. The big number variability between the groups can affect the results and the significance of the data.
(2) The experimental design report 4 groups of treatments but in the relative results section (3.1) were discussed and represented the data relative to 3 groups only. More important, the number of the animals mentioned in this section is different from the number present in the experimental design. Please explain the motivations.
(3) Figure 2C is very confusing for the reader. Only 3 of 4 groups (please insert Sham-MSC-EVs group) are reported and the symbols near each column are not necessary. If I well understand they represent the number of samples per group, but this information could be found in the figure legend.
(4) The reported data are relative to small groups and the experiment has been performed only one time. I know this is a complex model but the analyses don’t revealed any statistical significance so it will be better to underline that are “preliminary” data and the usage of “strong” conclusion is discarded (increased, importantly etc).
(5) Result 3.1 must be edited considering the previous indications and more attention could be placed in the comparison between groups, percentage and significance discussion.

2- in vitro experiments
(1) In vitro BBB models are largely used to perform integrity and permeability experiments. More groups use TEER values as set up to test cell monolayers integrity before experiments start.
In this work 150Ω has been chosen as baseline TEER values, however from a medline I have observed that many groups that work with this model (ECs cells) use higher TEER baseline value 200±50Ω for 6-well. Please indicate well plate used in these experiments and 150Ω justification.
(2) Please provide the formula used to calculate TEER value reported for treated groups
(3) In figure legend 3 please indicate that * reported in the graphs e and h correspond to p<0.05.
(4) No data are supplied about protein concentration of samples used for western blot and experiment replicates.

Minor revision
I could spot some minor typos and grammatical English errors (though I'm not a native English speaker).
Please in figure 3 g specify the KDa of M.
Please for all in vitro data indicate how many times the experiments have been performed

Author Response

We thank the reviewer for her/his interest in our work and appreciation of our new insights into the role of the BBB around HIE. Please find enclosed a point to point response to the comments of the second reviewer.

Round 2

Reviewer 2 Report

The manuscript has been improved in content and understanding and is acceptable for publication